# Cortical Surface Diffusion Generative Models

**Zhenshan Xie**[1]                    ZHENSHAN.XIE@KCL.AC.UK

**Simon Dahan** [1]                     SIMON.DAHAN@KCL.AC.UK

**Logan Z. J. Williams**[1]                LOGAN.WILLIAMS@KCL.AC.UK

**M. Jorge Cardoso**[1]                M.JORGE.CARDOSO@KCL.AC.UK

**Emma C. Robinson**[1]                EMMA.ROBINSON@KCL.AC.UK

[1]*Department of Biomedical Engineering, School of Biomedical Imaging &Imaging Sciences, King's College London, London, UK*

## Abstract

Cortical surface analysis is gaining prominence as a sensitive tool for studying complex neuropsychiatric disorders. However, patterns of cortical organization are complex and highly variable across individuals, challenging classical approaches for analysis that rely on diffeomorphic image registration. This leads to an urgent need for better models of brain development and diverse variability inherent across individuals. Traditional vision diffusion models have shown effectiveness in generating high-resolution and realistic natural images, which makes them particularly suited to addressing cortical surface problems, where the features of interest are subtle and highly variable across individuals. In this work, we first proposed a novel diffusion model for generating cortical surface metrics, using modified surface vision transformers as the principal architecture. We validate our method in the developing Human Connectome Project (dHCP) with results suggesting that our model demonstrates excellent performance in capturing the intricate details of evolving cortical surfaces - generating high-quality realistic samples of cortical surfaces conditioned on postmenstrual age (PMA) at scan.

**Keywords:** Cortical Surfaces, Diffusion Models, Generative Models

## 1. Introduction

The human cerebral cortex is a highly convoluted sheet of grey matter that is best modeled as a surface (Fischl et al., 1999; Robinson et al., 2014; Glasser et al., 2016b). As one of the most highly evolved areas of the human brain, cortical function underpins many aspects of higher-order cognition and this was implicated in many neurological and psychiatric disorders (Paus et al., 2008; Roe et al., 2021). Unfortunately, localizing signs of pathology in individual brains can be extremely challenging due to the scale of variation of human cortical organization and limited availability of data (Glasser et al., 2016a; Kong et al., 2019; Gordon et al., 2017). Therefore, it is desirable to develop methods that can help enrich the diversity of the variation from pathology or even conditioned on desired information.

To address this problem, we propose a diffusion probabilistic model on cortical surface based on a modified surface vision transformer (SiT) (Dahan et al., 2022) which has shown better ability to fully capture the feature of cortical organization across individuals compared to traditional convolutional networks. In this paper, we demonstrate that, with the modified SiT as a backbone for Denoising Diffusion Probabilistic Models (DDPM) (Ho et al., 2020), it is possible to conditionally generate feature maps that accurately simulate cortical neurodevelopment.

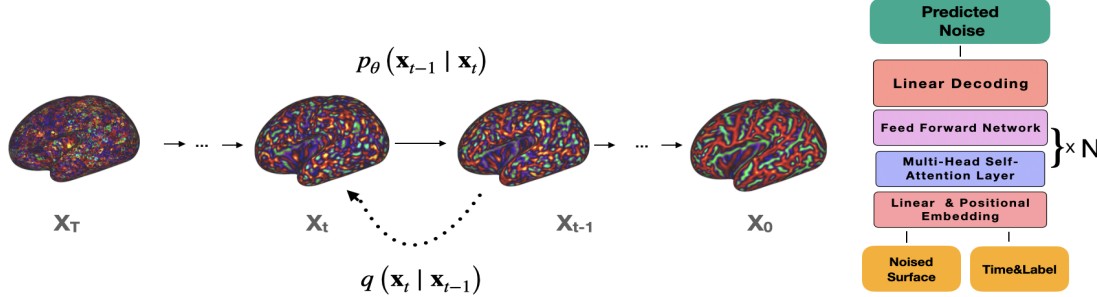

Figure 1: The image shows the directed graphical model of the diffusion process on cortical surface and the model architecture.

## 2. Methods

Following the DDPM formulation, diffusion models employ a structured generative process where a simple noise distribution is gradually transformed into complex data distributions through a series of iterative steps, guided by neural network architecture, that learns to reverse the diffusion process at each iteration, parameterized by a time variable $t$. In this paper, we first use the modified SiT to predict the noise. Samples $x_0 \in \mathbb{R}^{40962 \times 1}$ represent cortical curvature maps, sampled over a regular, sixth-order icospheric grid (with 40962 regularly spaced vertices).

**Encoding:** Following the setting of SiT, the encoding works by patching the high-resolution cortical feature maps, using faces from a second-order icosahedron (ico2-152 vertices, 320 patches). Patches are first projected onto a D-dimensional sequence of tokens, using a trainable linear layer. Positional embeddings are then added to encode spatial information about the sequence of tokens. Data is then passed through $N$ vision transformer blocks, each composed of a multi-head self-attention layer and a feed-forward network.

**Decoding:** To support the prediction of noise and diagonal covariance at the same resolution as the original data at each iteration, we then extend the network architecture by incorporating a layer normalization (adaptive if using adaLN) and a standard linear layer for decoding each token into $N \times (2V_p)$. This adaptation ensures that the data is precisely reprojected back to the original resolution of the patches. Finally, the decoded tokens are rearranged into their original spatial layout to regenerate surface maps.

**Condition Generation:** At the same time, we seek to further condition the model on the postmenstrual age at scan (PMA) of our samples ($a$) by appending $a$ and $t$ as additional tokens in the input sequence. We use classifier-free guidance to encourage the sampling procedure to find $x$ such that $\log p(c \mid x)$ is high.

## 3. Experiments

**Data:** We used 530 neonatal T2w MRI scans from the developing Human Connectome Project (dHCP): 419 term-born neonates (born after 37 weeks gestation) and 111 preterm neonates (born before 37 weeks),covering postmenstrual age (PMA) at scan between 24 to 45

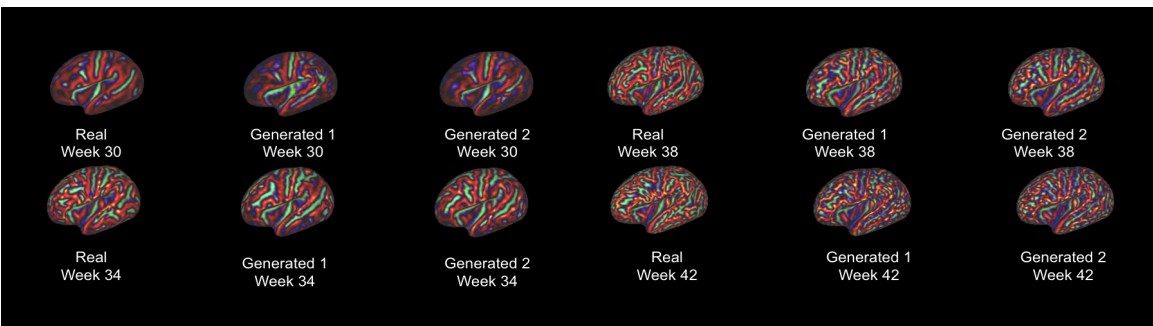

Figure 2: Samples of cortical surfaces generated by surface diffusion model conditioned on birth age of 30, 34, 38, and 42 weeks, compared to the real samples in dHCP dataset

weeks. Cortical surface processing was performed using the dHCP structural pipeline (Makropoulos et al., 2018). In this experiment, we use curvature maps, projected to the sphere. Training, test, and validation sets were allocated in the ratio of 423:54:53.

**Training** The transformer backbone was implemented following the Tiny-SiT setting in (Dahan et al., 2022). The training was implemented with a constant learning rate of $1 \times 10^{-4}$, and a batch size of 180 on NVIDIA 3090 with 24 GB memory.

**Evaluation of Conditional Generation** The model was validated through visual inspection and training of an independent SiT regression model which evaluated whether the generated samples convincingly represent curvature maps of a specific PMA. This regression model was trained on the same data set used for generation, with PMA as the target variable. The proposed surface DDPM was used to generate 20 synthetic examples for each week, from 27 weeks to 44 weeks. The results in table 3 report mean absolute error for PMA predictions obtained on ground truth testing data and on synthetic data. Visual examples are shown in Fig 2.

|  | Ground Truth | Synthetic |
|---|---|---|
| MAE | $0.59 \pm 0.51$ | $0.80 \pm 0.50$ |
| r2 | 0.96 | 0.96 |

Table 1: Prediction results on dHCP test and synthetic data for the PMA regression task. Mean Absolute Error with standard deviation and $R^2$ score are reported.

## 4. Conclusion

This paper presents a general cortical surface diffusion generative model validated on the conditioned generation of cortical curvature maps across neurodevelopment. The generated samples demonstrate that surface diffusion can capture expressive features on the surface sufficiently well to sensitively model the rapid changes in cortical curvature that occur across late gestation.

## Acknowledgments

We gratefully acknowledge E.C.R. supported by an Academy of Medical Sciences Springboard Award (SBF003/1116), an MRC Methodology Research grant (MR/V03832X/1) and King's-China Scholarship(K-CSC) PhD Scholarship program. Data were provided by the developing Human Connectome Project. We are grateful to the families who generously supported this trial.

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
