# OpenReview forum: "Cortical Surface Diffusion Generative Models"
_MIDL.io/2024/Short_Papers — MIDL 2024 Short Papers_

### Official Review · Reviewer_hrnV · 2024-04-24

**Confidence:** 5
**Final Rating:** 4

**Review:**

This paper introduces a denoising diffusion probabilistic models (DDPM)-based generative model for synthesizing cortical surfaces, which is of an increasing interest in the medical domain. The authors validate the generated samples using the HPC dataset.

Pros:

The paper is well-written and easy to follow. The proposed method is straightforward as it is an application of DDPM.

Cons:

The evaluation could be strengthened by considering other data metrics, such as FID, to validate the fidelity and reliability of the generated samples.

It would be beneficial if the authors could provide insights into why the R^2 statistics (in Tab. 1) are the same, while the mean absolute error obtained on ground truth testing data and on synthetic data differs significantly.

---

### Decision · Program_Chairs · 2024-04-26

Accept